# The Quest for the Right Mediator: A History, Survey, and Theoretical Grounding of Causal Interpretability

## Abstract

Interpretability provides a toolset for understanding how and why neural networks behave in certain ways. However, there is little unity in the field: most studies employ ad-hoc evaluations and do not share theoretical foundations, making it difficult to measure progress and compare the pros and cons of different techniques. Furthermore, while mechanistic understanding is frequently discussed, the basic causal units underlying these mechanisms are often not explicitly defined. In this paper, we propose a perspective on interpretability research grounded in causal mediation analysis. Specifically, we describe the history and current state of interpretability taxonomized according to the types of causal units (mediators) employed, as well as methods used to search over mediators. We discuss the pros and cons of each mediator, providing insights as to when particular kinds of mediators and search methods are most appropriate depending on the goals of a given study. We argue that this framing yields a more cohesive narrative of the field, as well as actionable insights for future work. Specifically, we recommend a focus on discovering new mediators with better trade-offs between human-interpretability and compute-efficiency, and which can uncover more sophisticated abstractions from neural networks than the primarily linear mediators employed in current work. We also argue for more standardized evaluations that enable principled comparisons across mediator types, such that we can better understand when particular causal units are better suited to particular use cases.

## 1 Introduction

To understand how neural networks (NNs) generalize, we must understand the causes of their behavior. These causes include inputs, but also the intermediate computations of the network. How can we understand what these computations represent, such that we can arrive at a deeper algorithmic understanding of how and why models behave the way they do? For example, if a model decides to refuse a user's request, was the refusal mediated by an underlying concept of toxicity, or by the presence of superficial correlates of toxicity (such as the mention of particular demographic groups)? The former would be significantly more likely to robustly and safely generalize. These questions motivate the field of causal interpretability, where we aim to extract causal graphs explaining how intermediate NN computations mediate model outputs.

This survey takes an opinionated stance on interpretability research: we ground the state of the field through the lens of causal mediation analysis (§2). We start by presenting a history of causal interpretability for neural networks (§3), from backpropagation (Rumelhart et al., 1986) to the beginning of the current causal and mechanistic interpretability wave.

We then survey common mediators (units of causal analysis) used in causal interpretability studies (§4), discussing the pros and cons of each mediator type. Should one analyze individual neurons? Full activation vectors? Model subgraphs? More broadly: **what is the right unit of abstraction for analyzing and discussing neural network behaviors?** Any NN component has pros and cons related to its level of granularity, whether it is a causal bottleneck, and whether it is natively part of the model (as opposed to whether it is learned on top of the model). After discussing common mediator types, we then categorize and discuss methods for searching over mediators of a given type to find those that are causally relevant to some

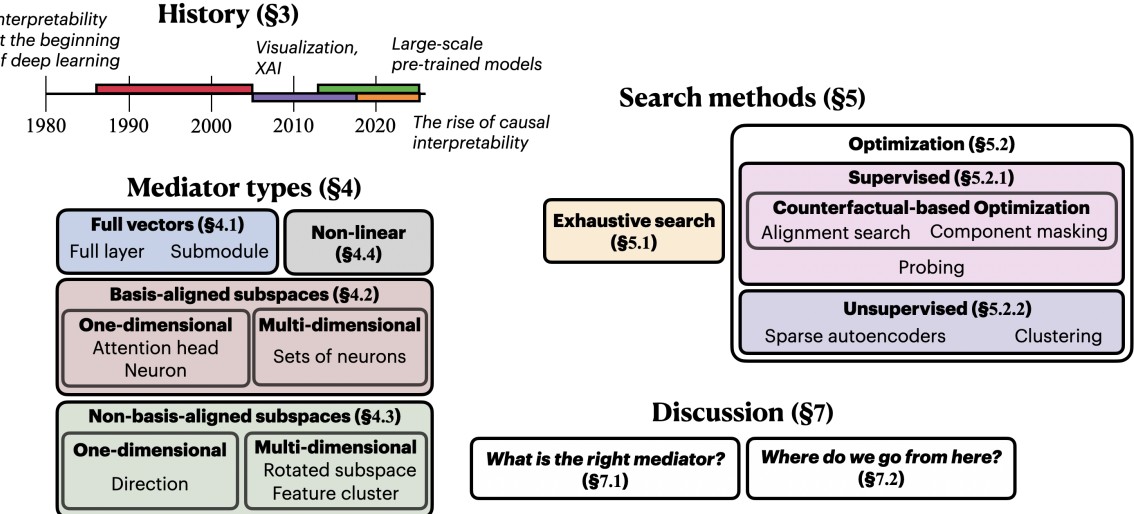

Figure 1: Outline of survey. After defining necessary causal terminology (§2), we give an overview of the history of causal interpretability centered on units of causal analysis (§3). We then survey and categorize commonly used units of analysis and describe their strengths and weaknesses (§4), as well as methods for searching over them (§5). After contextualizing our perspective with others' (§6), we discuss (§7) what we consider to be among the most important questions in causal interpretability: **What is the right unit of causal analysis (§7.1)? Given this perspective, what kinds of mediators and research will be needed to advance the field (§7.2)?**

task (§5). Finally, after surveying the field (§6), we (1) point out mediators which have been underexplored, but have significant potential to yield new insights; (2) propose criteria that future mediators should satisfy, based on the goals of one's study; and (3) suggest ways to measure progress in causal interpretability moving forward (§7). Figure 1 summarizes the content and flow of this paper.

## 2 Preliminaries

**The counterfactual theory of causality.** Lewis (1973) poses that a **causal dependence** holds iff the following condition holds:

> "An event $E$ *causally depends* on $C$ [iff] (i) if $C$ had occurred, then $E$ would have occurred, and (ii) if $C$ had not occurred, then $E$ would not have occurred."

Lewis (1986) extends this definition of causal dependence to be whether there is a **causal chain** linking $C$ to $E$; a causal chain is a connected series of causes and effects that proceeds from an initial event to a final one, with potentially many intermediate events between them. This idea was later extended from a binary notion of whether the effect happens to a more nuanced notion of causes having influence on *how* or *when* events occur (Lewis, 2000). Other work defines notions of cause and effect as continuous measurable quantities (Pearl, 2000); this includes direct and indirect effects (Robins & Greenland, 1992; Pearl, 2001), which are common metrics in causal interpretability studies.

**Causal abstractions in neural network interpretability.** Causal interpretability is based on the abstraction of **causal graphs**. A causal graph is a graphical model consisting of nodes $V$ and directed edges $E$. A **node** in a causal graph corresponds to an action or event; in neural networks, it can correspond to any component (or combination thereof), as described below and in §4. An **edge** encodes a causal relationship, where the source is the cause and destination is the effect. For example, if edge $e$ is drawn from one neuron

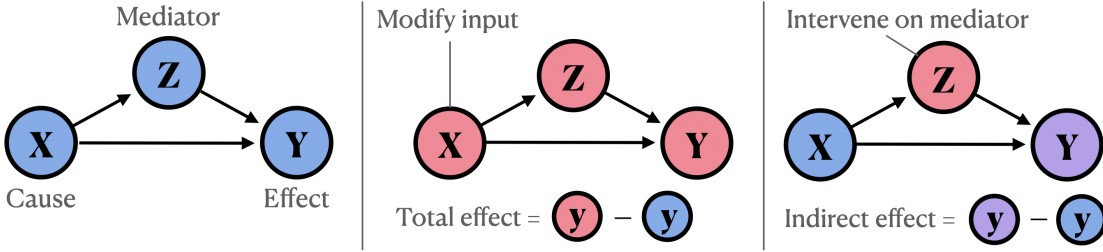

Figure 2: Visual summary of causal mediation analysis. Given a causal graph from inputs $X$ to outputs $Y$ and a specific cause (input) $X = x$, we can observe a resulting effect (e.g., a model behavior) $Y = y$. There often exist variables such as $Z$ that mediate the influence of $X$ on $Y$. A common way to quantify the importance of $Z$ is by measuring its **indirect effect** (Eq. 1), where, given $X = x$, one intervenes on the value of $Z$ and then measures the change in $y$ relative to when no intervention was performed.

to another, this indicates that the first neuron has strong counterfactual influence over the second.[1] It is also frequently required that $e$ have strong influence on the final target behavior, rather than just the downstream intermediate component. In the case of neural networks, a node can be any unit or intermediate representation produced by the network: it could be a neuron, a full layer, an attention head, or even some combination of these. An edge is more abstract: it encodes a causal relationship between any two nodes in the network. The only restriction is that the source of the edge come before the destination in the computation graph. If we are given an input cause $X = x$ and output effect $Y = y$, there may exist many causal nodes between them; these intermediate nodes are called **mediators**.

Any node in a causal graph can function as both a cause and an effect. All edges have a source node and destination node; a **cause** is a source node, and an **effect** is a destination node. Therefore, in a neural network, there are many intermediate causes and effects that explain how inputs (the original cause of a behavior and first node of the causal graph) are transformed into model behaviors (the model behavior and final node of the causal graph).

In the causality literature, a **mechanism** is defined as a causal chain from cause $C$ to effect $E$. The mechanistic interpretability literature, while closely related to causal interpretability, does not enforce this causally-grounded definition of mechanism: mechanistic interpretability is often defined as reverse-engineering neural networks to better understand how and why they behave in certain ways (cf. Miller et al., 2024; Nanda et al., 2023). The overlap between mechanistic and causal interpretability is significant, but not total: for example, sparse autoencoders (Bricken et al., 2023; Cunningham et al., 2024) are correlational, but many methods, such as circuit discovery (Elhage et al., 2021; Conmy et al., 2023) and alignment search (Geiger et al., 2021; 2024), employ methods to causally implicate model components (or other abstractions discovered in potentially correlational ways) in model behavior. We believe that the causality-based definition of mechanism is a useful one that makes precise the main challenge of mechanistic interpretability—to reverse-engineer an algorithmic understanding of neural network behaviors, where "algorithm" is essentially equivalent to a complete causal graph explaining how a model will generalize.

The abstraction of causal graphs extends naturally to neural networks: we can treat the computation graph of a neural network as the full causal graph which explains how inputs $X$ are transformed into a probability distribution over outputs $Y$. In this case, all model components[2] can be viewed as causal nodes that mediate the transformation of $X$ to $Y$. This is discussed in detail in §4.

**Counterfactual interventions.** In interpretability, "causal method" generally refers to a method that employs **counterfactual interventions** (Lewis, 1973) to some part of the model or its inputs. Much early

---

[1]This section focuses on defining causal graph terminology. See §4 for definitions of neural network component types like neurons and submodules.

[2]We will use "component" primarily to refer to neurons and attention heads. This is imprecise, but captures atomic units of the computation graph that are often used as mediators in current work. It also serves to contrast with mediators that cannot be easily extracted from the computation graph, such as non-basis-aligned directions.

interpretability work focused on interpreting model decision boundaries by intervening on the inputs, but contemporary work is primarily concerned with understanding which intermediary model components are responsible for some behavior given some input—i.e., finding the right nodes from the low-level computation graph to keep in the high-level causal graph.

Causal mediation analysis (Pearl, 2001) provides a unified framework for performing counterfactual interventions. Given input $X$, an output $Y$, and a causal graph consisting of many intermediate nodes betwen $X$ and $Y$, the causal influence of an intermediate node (mediator) $Z$ on a downstream node $Y$ is quantified as $Z$'s **indirect effect** (IE; Pearl, 2001; Robins & Greenland, 1992). This metric is based on the notion of counterfactual dependence, where one measures the difference in some output metric $m$ before and after intervening on a given mediator $Z$; $m$ is generally derived from $Y$. It is common to measure $m$ given a normal run of the model on input $X = x$, where $Z$ takes its natural value $z$, and then compare this to $m$ given $X = x$ where mediator $Z$ is set to some alternate value $z'$:[3]

$$\text{IE}(m; X, x; Z, z, z') = m(X = x \mid Z = z) - m(X = x \mid \text{do}(Z = z'))\tag{1}$$

See Figure 2 for an illustration.

## 3 A History of Causal Interpretability

Causal interpretability techniques have existed since the beginning of deep learning. What distinguishes the current wave of mechanistic interpretability studies from past work in causal interpretability, and what lessons can past work (which often used very different methods and mediators to contemporary studies) teach us about analyzing intermediate model computations? We claim that the lens of causal mediation analysis enables a clear and novel narrative of the trajectory of interpretability research; links current issues in the field to longstanding issues that have been argued to exist since the 1980s; and highlights new and impactful types of open problems.

**Interpretability at the beginning of deep learning.** In 1986, Rumelhart et al. published an algorithm for backpropagation and an analysis of this algorithm. This enabled and massively popularized research in training multi-layer perceptrons (MLPs)—now often called feedforward layers. This paper arguably represents the first mechanistic interpretability study: the authors evaluated their method by inspecting each activation and weight in the neural network, and observing whether the learned algorithm corresponded to the human intuition of how the task should be performed. In other words, they reverse-engineered the algorithm of the network by labeling the rules encoded by each **neuron** and **weight**!

Throughout the 1990s and early 2000s, the idea of extracting rules by analyzing each weight and activation at the neuron level remained popular. At first, this was a manual process: networks were either small enough to be manually interpreted (Rumelhart et al., 1986; McClelland & Rumelhart, 1985) or interpreted with the aid of carefully crafted datasets (Elman, 1989; 1990; 1991); alternatively, researchers could prune them (Mozer & Smolensky, 1988; Karnin, 1990) to a sufficiently small size to be manually interpretable. Later, researchers proposed techniques for automatically extracting rules (Hayashi, 1990) or decision trees from NNs (Craven & Shavlik, 1994; 1995; Krishnan et al., 1999; Boz, 2002)—often after the network had been pruned. At this point, interest in causal methods based on interventions had not yet been established, as networks were often small and/or simple enough to directly understand without significant abstraction. Nonetheless, as the size of neural networks scaled up, the number of rules encoded in a network increased; thus, rule/decision tree extraction techniques could not generate easily human-interpretable explanations or algorithmic abstractions of model behaviors beyond a certain size. This led to the rise of **visualization methods** in the 2000s, which became a popular way to demonstrate the complexity of phenomena that models had learned to encode. Designing visualizations of inputs and outputs of the network (Tzeng & Ma, 2005) and interactive visualizations of model activations (Erhan et al., 2009) were valuable initial tools for generating hypotheses as to what kinds of concepts models could represent. While visualization research was generally not causal, this subfield would remain influential for interpretability research as neural networks scaled in size in the following decade.

---

[3] Appendix A surveys methods for sourcing $z'$. This can come from alternate inputs where the answer is flipped, means over many inputs, or arbitrary constants (typically 0).

**Large-scale pre-trained models.** The 2010s were a time of rapid change in machine learning. In 2012, the first large-scale pre-trained neural network, AlexNet (Krizhevsky et al., 2012), was released. Not long after, pre-trained word embeddings (Mikolov et al., 2013a;b; Pennington et al., 2014) became common in natural language processing (NLP), and further pre-trained deep networks followed (He et al., 2016). These were based on ideas from *deep learning*. This represented a significant paradigm shift: formerly, each study would build ad-hoc models which were not shared across studies, but which were generally more transparent.[4] After 2012, there was a transition toward using a shared collection of significantly larger and more capable—but also more opaque—models. This raised new questions on what was encoded in the representations of these shared scientific artifacts. The rapid scaling of these models rendered old neuron-level rule extraction methods either intractable or made its results difficult to interpret; thus, interpretability methods in the early 2010s tended to prominently feature scalable and relatively fast *correlational* methods, including visualizations (Zeiler & Fergus, 2014) and saliency maps (Simonyan et al., 2014). This trend continued into 2014–2015, when recurrent neural network–based (Elman, 1990) language models (Mikolov et al., 2010) began to overtake statistical models in performance (Bahdanau et al., 2015); for example, visualizing RNN and LSTM (Hochreiter & Schmidhuber, 1997) hidden states was proposed as a way to better understand their incremental processing (Karpathy et al., 2016; Strobelt et al., 2017).

At the same time, interpretability methods started to focus more on *explaining* model predictions.[5] The explainable AI (XAI) field was and is extensive. One line of work designed supervised auxiliary (correlational) models to explain particular model predictions, such as LIME (Ribeiro et al., 2016a;b), Anchors (Ribeiro et al., 2018), and extensions like CLEAR that explicitly integrate notions of counterfactual fidelity to the output explanations (White & d'Avila Garcez, 2020). These models learn local decision boundaries, or some human-interpretable simplified representation of a model's behavior. Other works interpreted predictions via feature importance measures like SHAP (Lundberg & Lee, 2017). Influence functions (Koh & Liang, 2017) traced the model's behavior back to specific instances from the training data. Another line of work sought to directly manipulate intermediate concepts to control model behavior at test time (Koh et al., 2020), or to decompose distributed representations into interpretable symbolic representations post hoc (Odense & Garcez, 2020). The primary difference between these visualization-/correlation-/input-based methods and current methods is that these methods generally prioritize explaining high-level patterns about responses to particular kinds of inputs, such that we can generate hypotheses as to the types of *input concepts* or *intermediate rules* that explain particular model predictions. In contrast, current work prioritizes highly localized and causal explanations of *how* and in *which regions of the computation graph* models translate particular inputs into general output behaviors.

2017–2019 featured perhaps the largest architectural shift (among many) in machine learning methods at this time: Transformers (Vaswani et al., 2017) were released and quickly became popular due to scalability and high performance. This led directly to the first successful large-scale pretrained language models, such as (Ro)BERT(a) (Devlin et al., 2019; Liu et al., 2019b) and GPT-2 (Radford et al., 2019). These significantly outperformed prior models, though it was unclear why—and at this scale, analyzing neural networks at the neuron level using past techniques had long become intractable. This combination of high performance and little mechanistic understanding created demand for interpretability techniques that allowed us to see *how* language models had learned to perform so well. Hence, correlational probing methods rose to meet this demand: classifiers are trained on intermediate activations to extract some target phenomenon. Probing classifiers were used to investigate the latent morphosyntactic structures encoded in static word embeddings (Köhn, 2015; Gupta et al., 2015) or intermediate hidden representations in pre-trained language models—for example, in neural machine translation systems (Shi et al., 2016; Belinkov et al., 2017; Conneau et al., 2018) and pre-trained language models (Hewitt & Manning, 2019; Hewitt et al., 2021; Lakretz et al., 2019; 2021). However, probing classifiers lack consistent baselines, and the claims made in these studies were not often causally verified (Belinkov, 2021). For instance, although an intervention may target a task mapping of

---

[4]Many systems built before deep learning were based on feature engineering, and so the information they relied on was more transparent than in current systems.

[5]There has classically been a distinction between *local* and *global* interpretability; local interpretability is concerned with explaining specific model predictions, whereas global interpretability is concerned with explaining a given model behavior in general across examples or building inherently interpretable models (Guidotti et al., 2018). Causal mediation analysis can encompass both of these styles of interpretability. As recent work has tended to focus more on global interpretability, we devote more attention to this style of work, though we cite and acknowledge examples of local interpretability methods in this section.

$A \rightarrow B$, an alternative property $C$ may be predicted, which can potentially impede causal claims about $A \rightarrow B$ (Ravichander et al., 2021). This likely encouraged researchers to search for more causally efficacious methods.

**The rise of causal interpretability.** 2017–2018 featured the first hints of our current wave of causal interpretability, with research that directly investigated intervening on neurons. For example, Giulianelli et al. (2018) trained a probing classifier, but then used gradients from the probe to modify the activations of the network. Other studies analyzed the functional role of individual neurons in static word embeddings (Li et al., 2017) or latent representations of generative adversarial networks (GANs; Goodfellow et al., 2014) by forcing certain neurons on or off (Bau et al., 2019b). The idea of manipulating neurons to steer behaviors was then applied to downstream task settings, such as machine translation (Bau et al., 2019a). The field was more widely popularized in 2020, when Vig et al. (2020) proposed a method for assigning causal importance scores to neurons and attention heads. It was an application of the counterfactual theory of causality (Lewis, 1973; 1986), as well as Pearl's operationalization and measurements of individual causal nodes' effect sizes (Pearl, 2001; 2000). This encouraged a new line of work that aimed to faithfully localize model behaviors to specific components, such as neurons or attention heads—an idea that would become foundational to current causal and mechanistic interpretability research.

At the same time, however, researchers began to realize the significant performance improvements that could be gained by massively increasing the number of parameters and training corpus sizes of neural networks (Brown et al., 2020; Kaplan et al., 2020). Massively increasing model sizes resulted in more interesting subjects of study, but also rendered causal interpretability significantly more difficult just as its popularity began. Thus, a primary challenge of causal interpretability has been to balance the often-contradictory goals of (i) obtaining a causally efficacious understanding of how and why models behave in a given manner, while also (ii) designing methods that are efficient enough to scale to ever-larger models.

Presently, there exist many subfields of interpretability that propose and apply causal methods to understand which model components contribute to an observed model behavior (e.g., Elhage et al., 2021; Geiger et al., 2021; Conmy et al., 2023). Recently, there have also been efforts to discover more human-interpretable mediators by moving toward latent-space structures aside from (collections of) neurons (Cunningham et al., 2024; Bricken et al., 2023; Wu et al., 2023). These methods and their applications are the focus of the survey that follows.

**The lens of causal mediation analysis.** For much of the history of deep learning, neurons (§4.2) and layers (§4.1) were the basic unit of study in causal interpretability. They are natural units of the model, and in small systems, they can sometimes be human-interpretable. Large-scale pre-trained models have changed the field: neurons therein are numerous and generally *not* interpretable, so the field has now turned to more sophisticated abstractions like **sets of neurons**, **attention heads** (§4.2), or even **non-basis-aligned subspaces** (§4.3) that require external modules (such as probes or sparse autoencoders) to locate (§5.2). Different units of analysis have significantly different strengths and weaknesses, and the unit of analysis determines, to a large extent, the kinds of features that can be found, and the class of methods that can be used to find them (§5). In the following section, we more precisely define these units of analysis, and compare their strengths and weaknesses.

## 4 Selecting a mediator type

In this section, we discuss different types of causal mediators in neural networks, and the pros and cons of each. Figure 3 visualizes a computation graph, and units thereof that are often used as mediators in causal interpretability studies. In causal interpretability, we often do not want to treat the full computation graph as the final causal graph, as it is large and difficult to directly interpret. Thus, we typically want to build higher-level causal abstractions that capture only the most important mediators, and/or where each causal node is human-interpretable. In this section, our primary questions are the following: What kinds of model components can be used as mediators? What are the strengths and weaknesses of using particular kinds of components as mediators? Table 1 summarizes mediator types, their strengths and weaknesses, and search methods (§5) commonly used with each.

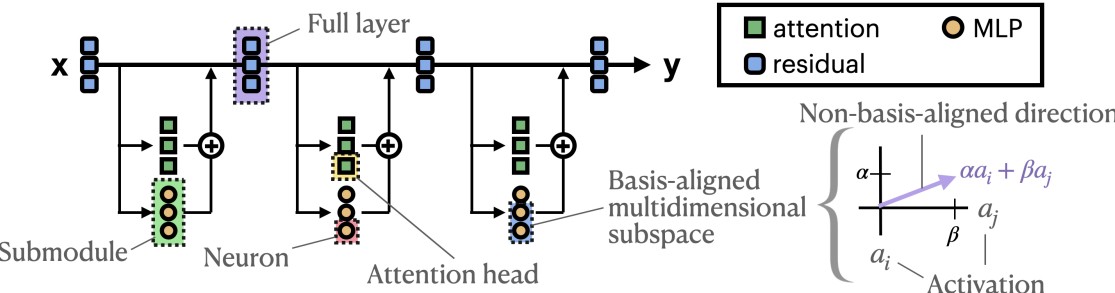

Figure 3: Visualization of common mediator types in neural networks. **Neurons** or **attention heads** are natural and common units of analysis. **Full layer** and **submodule** vectors are more coarse-grained, but more easily enumerable. One can also implicate a **multidimensional subspace**, which could be neuron-basis-aligned (as in a group of neurons, pictured here) or non-basis-aligned. Non-basis-aligned mediators—e.g., non-basis-aligned **directions**—have recently become a popular mediator type due to their monosemanticity. However, discovering non-basis-aligned mediators requires external modules such as classifiers, autoencoders, or other modifications to the original computation graph. Note that while this figure depicts a Transformer, many of the mediator types generalize to other architectures (the primary exception being attention heads).

Table 1: Summary of mediator types, the pros and cons of each, the search methods that are typically used to search over them, and examples of studies that employ them.

| Mediator type | Strengths | Weaknesses | Common search methods | Example studies |
|---|---|---|---|---|
| Full layers and submodules (§4.1) | Small search space. Some useful applications. | Difficult to interpret. Not well-suited to explaining model behavior. | Exhaustive search (§5.1), supervised probing (§5.2.1) | Hupkes et al. (2017); Giulianelli et al. (2018); Hewitt & Manning (2019); Geva et al. (2023); Meng et al. (2022) |
| Neurons and attention heads (§4.2) | Discrete and enumerable. Relatively fine-grained; sometimes enables control over NN behaviors. | Search space not always tractable. Often not interpretable. | Exhaustive search (§5.1) | Vig et al. (2020); Bau et al. (2020); Finlayson et al. (2021); Lakretz et al. (2019); Cao et al. (2021) |
| Non-basis-aligned spaces (§4.3) | Fine-grained. Interpretable. Enables precise control of NN behaviors. | Non-enumerable. Typically requires optimization; sensitive to training setup and random variance. May not be faithful to original model. | Optimization (§5.2): supervised probing (§5.2.1), unsupervised methods such as sparse autoencoders (§5.2.2) | Wu et al. (2023); Bricken et al. (2023); Cunningham et al. (2024); Marks & Tegmark (2023); Marks et al. (2024); Ravfogel et al. (2021) |

One possible mediator type is a **full layer**—typically the *output activations of* a specific layer $\ell$ (§4.1). This generally refers to an activation vector $\mathbf{a}^\ell$, where each index $\mathbf{a}^\ell_i$ is a **neuron** that can take some activation $a^\ell_i$.[6] One can also use the output vector of an intermediate **submodule** within the layer (e.g., an MLP), rather than the output of the whole layer. For example, in Transformers (Vaswani et al., 2017),[7] a layer typically consists of two submodules: a multi-layer perceptron (MLP) and an attention block, which can be arranged either sequentially or in parallel. The output of these submodules is also a vector of activations, so we will refer to their individual dimensions as neurons as well.[8]

One can also use single neurons or sets of neurons as a mediator (§4.2). If we use a set of neurons (possibly of size 1) $\{\mathbf{a}^\ell_i, \mathbf{a}^\ell_j, \ldots\}$ from a vector $\mathbf{a}^\ell$, this is referred to as a **basis-aligned subspace** of $\mathbf{a}^\ell$. A one-dimensional basis-aligned subspace is equivalent to a neuron; we will use basis-aligned subspace primarily to refer to neuron groups of size > 1. Basis alignment is a key concept: values that align with neuron directions will be discoverable without any external modules appended to the original computation graph. For example, it is straightforward to exhaustively search over and intervene on single neurons; it is less tractable, but still

---

[6]In other words, we use "neuron" to refer to any basis-aligned direction in activation space.

[7]Transformers are currently the dominant architecture for vision and/or language modeling; as such, there is much more work on interpreting the decisions of models built with this architecture. However, our ideas are presented in a general way that will also apply with minor modifications to other neural-network-based architectures, such as recurrent neural networks (Mikolov et al., 2010) and state space models.

[8]Using the same notation emphasizes that these are mediators of the same level of granularity. However, we acknowledge that this obscures that neurons in different locations often encode different types of features.

theoretically possible, to enumerate all $2^n$ possible combinations of neurons without using any additional parameters.

However, causally relevant features are not guaranteed to be aligned with neurons in activation space; indeed, there are many cases where human-interpretable features correspond to subspaces that are not spanned by a (set of) neuron(s) (Elhage et al., 2022b; Bricken et al., 2023). Thus, in recent studies, it is common to study **non-basis-aligned** spaces (§4.3). Each channel of a non-basis-aligned subspace can be defined as weighted linear combination of neuron activations. For example, to obtain a non-basis-aligned **direction**,[9] we could learn coefficients $\alpha$ and $\beta$ to weight activations $a_i^\ell$ and $a_j^\ell$ (optionally with a bias term $b$):

$$d = \alpha \cdot a_i^\ell + \beta \cdot a_j^\ell + \ldots + b \tag{2}$$

Note that these new constants $\alpha$ and $\beta$ are *not* part of the original model's computation graph. This means that discovering non-basis-aligned directions often requires external modules that weight components from the computation graph in some way—e.g., classifiers or autoencoders (§5.2).

The primary trade-off between these mediator types is their granularity and quantity. This section proceeds in order of increasing granularity and quantity.

## 4.1 Full layers and submodules

Full layers and submodules are relatively coarse-grained mediators. Thus, they are a common starting point if one does not know where in a model a particular type of computation is happening. Early probing classifiers studied the information encoded in full layers (Shi et al., 2016; Hupkes et al., 2018; Belinkov et al., 2017; Conneau et al., 2018; Hewitt & Manning, 2019; Giulianelli et al., 2018), and recent studies that leverage classifiers as part of causal techniques still frequently do the same (e.g., Elazar et al., 2021; Marks & Tegmark, 2023; Li et al., 2023). This makes layers a natural mediator for exploratory interventions where the usage of more fine-grained mediators is infeasible, as in Conmy et al. (2023), or where broad characterizations of information flow are sufficient, as in Geva et al. (2023); Sharma et al. (2024). That said, it is rare to completely ablate a layer and then observe how this changes behavior; this has been done in a pruning study where the motivation was not interpretability (Sajjad et al., 2023), but this technique has potential to inform our understanding of which general model regions are more responsible for certain kinds of behaviors (e.g. Lad et al., 2024).

In some cases, coarse-grained mediators like these can inform methods for understanding factual recall in language models (Geva et al., 2023), or updating these factual associations (Meng et al., 2022; 2023). However, full layers encode many features and have many causal roles in a network, which makes it difficult to interpret how, exactly, relevant information is encoded in a layer (Conmy et al., 2023). Additionally, intervening on full layers or submodules often causes side effects outside the scope of the intervention (McGrath et al., 2023).

The primary advantage of using full layers as mediators is their small quantity and broad scope of information. This means that even slow or resource-intensive methods will generally be easy to apply to all layers. In some cases, this is enough granularity. However, an obvious disadvantage is that this mediator is generally opaque: even if we know that information is encoded in a layer somehow, it is unclear *precisely* how this information is encoded, composed, or used. Thus, layers and submodules have little explanatory power, and are better used as coarser starting points for later finer-grained investigations (e.g., Brinkmann et al., 2024; Geva et al., 2023) or for downstream applications such as model editing (Meng et al., 2022; 2023; Sharma et al., 2024; Gandikota et al., 2023; 2024).

## 4.2 Basis-aligned subspaces

**Neurons.** Compared to full layers and submodules, neurons represent more fine-grained components within neural networks that could feasibly represent individual features (though we discuss below that this is not often the case due to polysemanticity). Individual neurons can be considered the smallest meaningful unit

---

[9]We will use "direction" to refer to one-dimensional (sub)spaces.

within a neural network; an activation from a neuron is simply a scalar corresponding to a single dimension (1-dimensional subspace) of a hidden representation vector. Each neuron can differ from another based on its functional role in the network; for instance, Bau et al. (2020) locate neurons in a GAN responsible for generating specific types of objects in images, such as trees or windows, and verify this causally by ablating or artificially activating those neurons.

Neurons are a natural choice for mediator, as they are both fine-grained and easy to exhaustively iterate over (see §5.1). However, a major disadvantage of using neuron-based interpretability methods is *polysemanticity*. Individual neurons are often polysemantic—i.e. they respond to multiple seemingly unrelated inputs (Arora et al., 2018). For example, if the same neuron were sensitive to capitalized words, animal names, one-digit numbers, among other phenomena, it would be difficult to disentangle each of these individual patterns such that we can assign a coherent textual label to the neuron. Elhage et al. (2022b) investigate this phenomenon and suggest that neural networks represent features through linear superposition, where they represent features along non-basis-aligned linear subspaces, resulting in interpretable units being smeared across multiple neurons. In other words, in an activation vector of size $n$, a model can encode $k \gg n$ concepts as linear directions (Park et al., 2023), such that only a sparse subset of concepts are active given a particular input.

**Basis-aligned multidimensional subspaces.** The computations of individual neurons are not entirely independent: it may often be the case that *sets of* neurons compose to encode some concept. For example, in language models, localized subsets of neurons can be implicated in encoding gender bias (Vig et al., 2020), and implementing latent linguistic phenomena (Finlayson et al., 2021; Mueller et al., 2022; Bau et al., 2019a; Lakretz et al., 2019). Thus, some initial causal interpretability work employed heuristic-based searches over sets of neuron responsible for some behavior (e.g., Bau et al., 2019b; Vig et al., 2020; Cao et al., 2021). This is a generalization of individual neurons as mediators, where multiple dimensions in activation space are intervened upon simultaneously.

Using arbitrarily-sized sets of neurons gives us strictly more information, and thus potentially more descriptive mediators. Despite this, basis-aligned multidimensional subspaces are not commonly studied mediators. This is for two primary reasons: (1) There is a combinatorial explosion when we are allowed to search over arbitrarily-sized sets of neurons, which makes exhaustive searches intractable. (2) Additionally, interpretable concepts are not guaranteed to be aligned to neuron bases, meaning that leveraging groups of neurons still does not directly address the problem of polysemanticity—in fact, it may exacerbate the problem by adding even more information (Morcos et al., 2018; Chughtai et al., 2023; Wang et al., 2023).

**Attention heads.** Similar to neurons, attention heads are fundamental components of Transformer-based neural networks: they mediate the flow of information between token positions (Vaswani et al., 2017). Thus, using attention heads as units of causal analysis can help us understand how models synthesise contextual information (Ma et al., 2021; Neo et al., 2024) to predict subsequent tokens (Wang et al., 2023; Hanna et al., 2023; Prakash et al., 2024; García-Carrasco et al., 2024; Brinkmann et al., 2024). For practical purposes, each head within a layer can be understood as an independent operation, contributing a result that is then added into the residual stream.[10] For example, some heads specialise on syntactic relationships (Chen et al., 2024a), others on semantic relationships such as co-reference (Vig et al., 2020), and others still on maintaining long-range dependencies in text (Wu et al., 2024a). Attention heads have also been directly implicated in acquiring the ability to perform in-context learning (Olsson et al., 2022; Brown et al., 2020), or to detect and encode functions in latent space (Todd et al., 2024).

Attention heads are attractive mediators because they are easily enumerable (there are far fewer attention heads than neurons in a model) and because they often encode sophisticated multi-token relationships. However, in contrast to the activation of a neuron, the output of an attention head is multidimensional. Thus, it is difficult to directly interpret the full set of functional roles a single head might have; indeed, attention heads are almost always polysemantic, so one cannot typically determine the function(s) of an

---

[10]This is the *residual stream perspective* (Elhage et al., 2021) of Transformers, which has been adopted in recent interpretability research (Ferrando et al., 2024). The *residual stream perspective* suggests that the residual stream, which comprises the sum of the outputs of all the previous layers and the original input embedding, acts as a passive communication channel through which the MLP and attention submodules route the information they add.

attention head solely by observing its activations (Janiak et al., 2023)—as with neurons.[11] It has additionally been observed that ablating attention heads can cause other attention heads to compensate, which further complicates their analysis (Jermyn et al., 2023; Wang et al., 2023; McGrath et al., 2023).[12]

### 4.3 Non-basis-aligned spaces

**Non-basis-aligned multidimensional subspaces.**  Due to their polysemanticity, neurons, attention heads, and sets thereof do not necessarily correspond to cleanly interpretable features or concepts. For example, it is common that individual neurons activate on many seemingly unrelated inputs (Elhage et al., 2022b), and this issue cannot be cleanly resolved by adding more dimensions. This is because the features may actually be encoded in directions or subspaces that are *not aligned to neuron bases* (Mikolov et al., 2013a; Arora et al., 2016).

To overcome this disadvantage, one can generalize causal mediators to include arbitrary *non-neuron-basis-aligned* activation subspaces. This allows us to capture more sophisticated causal abstractions encoded in latent space, such as causal nodes corresponding to greater-than relationships (Wu et al., 2023), or equality relationships (Geiger et al., 2024). A common way of locating these is through learned rotation operations (Geiger et al., 2021; 2024), which preserve linearities and therefore are still in the activation subspace.

The primary advantage of considering an arbitrary subspace as a mediator is its expressivity: subspaces often capture distributed abstractions that are not fully captured by a single neuron. However, they are generally more difficult to locate than basis-aligned components, or non-basis-aligned directions, as we are typically required to have specific hypotheses as to how models accomplish a task, access to labeled data that isolates the target subspace, or enough compute to cluster existing mediators in an unsupervised manner. This is discussed in more detail in §5.2.

**Directions.**  A recent line of work aims to automatically identify specific directions (one-dimensional spaces) that correspond to monosemantic concept representations. Identifying and labeling these monosemantic model abstractions (often called *features*; Bricken et al., 2023; Cunningham et al., 2024; Huang et al., 2024) can reveal units of computation the model uses to solve tasks in a way that is often easier for humans to interpret.[13]

There is also initial evidence that these directions may enable fine-grained model control (Panickssery et al., 2024; Marks et al., 2024; Tigges et al., 2023). Past work has found initial signs that basis-aligned directions could be leveraged to edit (Meng et al., 2022) or steer (Turner et al., 2023; Paulo et al., 2024) model behavior, whereas more recent work has tended toward non-basis aligned directions. For example, there is work that uses linear probes to understand and ablate the effects of a direction on the model behavior Chen et al. (2024b); Ravfogel et al. (2020); Elazar et al. (2021); Ravfogel et al. (2021); Lasri et al. (2022); Marks & Tegmark (2023), as well as work that ablates (Marks et al., 2024; Cunningham et al., 2024) or injects (Templeton et al., 2024) directions corresponding to fine-grained concepts such as typically female names or the Golden Gate Bridge.

Nonetheless, directions still have key disadvantages. The search space over non-basis-aligned directions is infinite, making it impossible to exhaustively search over them. In fact, to discover these, we are generally *required* to modify the computation graph, as learning the coefficients on each neuron requires us to learn new parameters corresponding to the desired features. Regardless of the method used, each introduces confounds due to the stochastic optimization or significant manual effort required to locate these directions or subspaces.

---

[11]However, there is initial evidence that some dimensions of an attention head's output can be meaningfully explained (Merullo et al., 2024a;b). Thus, by decomposing the vector output of a head into smaller subspaces or even individual neurons, it may be easier to explain the set of functional roles of a given head.

[12]This phenomenon where downstream components only have causal relevance after an upstream component has been ablated is sometimes called **preemption** in the causality literature (Mueller, 2024). Preemption is not necessarily limited to attention heads; future work should thus analyze how common preemption is between other types of components, such as MLP submodules.

[13]Note that these directions are not necessarily subspaces of activation space: there are often non-linearities used in computing them, even though the vectors in activation space are involved in computing the directions. Therefore, we will refer to any 1-dimensional space as a **direction**, but do not require it to be a subspace of activation space.

### 4.4 Non-linear Mediators

Non-basis-aligned directions/subspaces are the most general *linear* mediator type. However, recent work has demonstrated that some features in language models can be represented non-linearly and/or using multiple dimensions. For example, there exist circular features representing days of the week or months of the year (Engels et al., 2024). Similarly, past work has found that many concepts can be more easily extracted using non-linear probes (Liu et al., 2019a), and that non-linear concept erasure techniques tend to outperform strictly linear techniques (Iskander et al., 2023; Ravfogel et al., 2022). However, in causal and mechanistic interpretability, most work has thus far tended toward using linear representations as units of causal analysis. Thus, there is significant potential in future work for systematically locating non-linearly-represented features—e.g., using group sparse autoencoders (Theodosis & Ba, 2023), which could isolate multiple directions simultaneously, and/or probing and clustering techniques to identify multidimensional features (Engels et al., 2024). Non-linear features have not been extensively studied, despite their expressivity; we therefore advocate investigating these mediators in §7.

## 5 Searching for task-relevant mediators

Once one has selected a task and a type of mediator, how does one identify task-relevant mediators of that type? The answer depends largely on the type of mediator chosen. If NNs have only a finite set of mediators of the chosen type—as is the case for native model components such as neurons, layers, and submodules—one could perform an exhaustive search over all possible mediators, choosing which to keep according to some metric; §5.1 discusses this approach. However, other mediator types, including non-basis-aligned directions and subspaces, carry a continuous space of possible mediators, rendering an exhaustive search impossible. A common solution to this problem is to employ optimization to either search this space or narrow the space into an enumerable discrete set, as discussed in §5.2.

### 5.1 Exhaustive search over mediators

Suppose we are given a neural network with a finite set of candidate mediators $\{Z_i\}_{i=1}^N$, such as the set of all neurons. One way to identify task-relevant mediators from this set is to assign each mediator $Z_i$ a task-relevancy score $S(Z_i)$ and then select the mediators with the top scores. This generally entails iterating over each candidate mediator $Z_i$, setting its activation to some counterfactual value (either from a different input where the answer is flipped, or a value that destroys the information within the neuron, such as its mean value), and then measuring how much this intervention changes the output. For example, Vig et al. (2020) and Finlayson et al. (2021) perform counterfactual interventions to the activation of each neuron individually and then quantify how much each neuron changes the probability of correct completions. The task relevancy score $S(Z_i)$ is typically the indirect effect (IE; Pearl, 2001; Robins & Greenland, 1992), as defined in Eq. 1.[14] This metric is based on the notion of counterfactual dependence, where we measure the difference in some output metric $m$ before and after intervening on a given component $Z_i$.

Exhaustive searches have many advantages: their results are comprehensive, causally efficacious, and relatively conceptually precise if our mediators are fine-grained units, like neurons. They are also open-ended, meaning that we are not required to have a pre-existing causal hypothesis as to how a model performs the task: we may simply ablate, artificially activate, or otherwise influence a component's activations, and then observe how it changes the output behavior or probability of some continuation. Due to these advantages, this method is the most common when we have a finite set of mediators—for example, in neuron-based analyses (Vig et al., 2020; Geiger et al., 2021; Finlayson et al., 2021) or attention-head-based analyses (Vig et al., 2020; Conmy et al., 2023; Syed et al., 2023).

However, exhaustive searches also have two significant disadvantages. The most obvious is that, in its exact form, an exhaustive search requires $O(N)$ forward passes, where $N$ is the number of mediators. This does not

---

[14]Other causal metrics include the **direct effect**, which measures the direct influence of the input on the output behavior except via the mediator. While more rarely used, it can be a helpful metric in tandem with indirect effects, as in Vig et al. (2020). There is also the **total effect**, which is the impact of changing the input on the model's output behavior. Note that the total effect does not directly implicate any particular component in model behavior, as it depends only on the input.

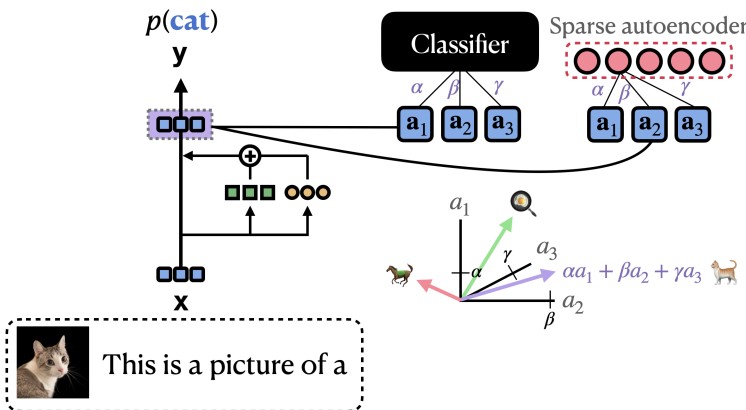

Figure 4: Neurons are not guaranteed to encode interpretable features. If non-basis-aligned directions encode the true features of interest, then a neuron may activate on many different features that are non-orthogonal to its corresponding basis. Locating non-basis-aligned mediators requires components in addition to the model's computation graph that encode the coefficients on each activation. For example, one can obtain these coefficients via supervised optimization with probing classifiers (§5.2.1) or unsupervised optimization with sparse autoencoders (§5.2.2). Note that optimization-based techniques sometimes introduce non-linearities, meaning that the discovered directions will not necessarily be a subspace of activation space.

scale efficiently as models scale; this is both because the number of components increases, but also because the computational cost of inference scales with model size. This may be why exhaustive searches have not often been extended to *sets of* neurons or heads, as this results in a combinatorial explosion in the size of the search space. Searches over sets of components can be approximated using greedy or top-k approaches, as in Vig et al. (2020), but this does not provide a comprehensive solution to the problem of assigning causal credit to groups of components. That said, there exist fast linear approximations to activation patching that are technically not causal and not always accurate, but that only require $O(1)$ forward and backward passes—most prominently, attribution patching (Kramár et al., 2024; Syed et al., 2023) and improved versions thereof inspired by integrated gradients (Sundararajan et al., 2017; Marks et al., 2024; Hanna et al., 2024).[15]

The second and more difficult disadvantage to overcome is that using exhaustive search constrains us to finite sets of mediators. Thus, this approach will not work well as-is if the search space is continuous (infinitely large). This is a key motivation behind the methods in the following subsection.

## 5.2 Optimizing over large spaces of mediators

For some types of mediators, the collection of candidate mediators is continuous or far too large to exhaustively search over; this precludes using methods described in §5.1. To search over large but enumerable sets, some researchers employ modified versions of exhaustive search, including greedy search methods (Vig et al., 2020) or manual searches (Wang et al., 2023). For continuous spaces, however, interpretability researchers generally use optimization. We taxonomize these optimization problems based on whether they require the interpretability researcher to manually select and incorporate task-specific information into the loss function (supervised methods; §5.2.1) or not (unsupervised methods; §5.2.2). We illustrate the intuition behind optimization-based search in Figure 4.

### 5.2.1 Supervised mediator searches

By *supervised mediator searches*, we mean parametric approaches which require labeled task data and/or human-generated hypothesized causal graphs. For example, these methods might require the researcher to

---

[15]Gradient-based methods are not causal because they do not directly establish counterfactual dependence. However, they do provide a scalar value whose magnitude can be interpreted as a local approximation of a model component's impact on the output.

propose candidate intermediate concepts which they expect the model to use in performing some task, or a candidate mechanism by which the model might complete the task. Others might simply require labeled data for training classifiers.

**Supervised probing.** In supervised probing approaches, the researcher proposes task-relevant concepts, and searches for mediators whose values are *correlated* with those concepts. Generally, one probes the hidden representation vector at the end of a particular layer, and the probe searches over all possible subspaces and directions therein for signals that are predictive of the labels. There exist many papers that employ probing classifiers (Belinkov & Glass, 2019), though many of them do not validate the causal efficacy of the probe's findings (Belinkov, 2021). A drawback of this is that NN units are often *correlated* with a concept without *causally mediating* the concept. Thus this approach can return many false positives—i.e. it may return proposed mediators which do not actually causally mediate the concept in question (Hewitt & Liang, 2019; Elazar et al., 2021; Amini et al., 2023; Belinkov, 2021).

Thus, much recent work complements supervised probing approaches with additional checks of causality—for example, by applying causal mediation analysis to the directions identified by supervised probing (Marks & Tegmark, 2023; Nanda et al., 2023); by backpropagating from the classifier to modify the behavior of the model (Giulianelli et al., 2018) or generate counterfactual representations (Tucker et al., 2021); or by directly comparing the probe's predictions to a causally grounded probe (Amini et al., 2023). Another line of work uses the directions discovered by probes to *guard* or *erase* information about a particular concept from the model's representations. For example, a direction in a model's activation space that is most predictive of the target concept can be nullified via orthogonal projections, such that the model can no longer use that information (Ravfogel et al., 2020); this process can then be repeated until linear guarding is achieved. Concept erasure and guarding can then be used to measure the causal importance of particular concepts, as in Elazar et al. (2021), though studies that employ methods like these tend to focus on single layers. More recently, techniques such as LEACE (Belrose et al., 2023) and follow-ups (Singh et al., 2024b) have generalized this idea to provably prevent any linear classifier from using a concept; this moves beyond orthogonal projections and projects out the information at *every* layer. One could use such methods to causally understand the *set of* directions that encode some concept. Note that these methods are still susceptible to the problems entailed by using linear mediators; thus, future work could extend erasure methods to non-linear mediator types.

**Counterfactual-based optimization.** Another class of approaches involves using the result of causal mediation analysis as a metric to directly optimize. One such line of work includes methods like Distributed Alignment Search (DAS) and follow-up methods such as Boundless DAS that align hypothesized high-level causal variables with underlying neural representations (Geiger et al., 2024; Wu et al., 2023; Huang et al., 2024). These methods decompose the search space into learnable subspaces that can be aligned with a hypothesized causal variable.

Another line of work learns a binary mask over enumerable sets of components to determine which are relevant mediators for a task, where the size of the set depends on the desired granularity (e.g. sets of neurons, attention heads, layers, etc.). Examples include subnetwork probing (Cao et al., 2021) and Desiderata-based Component Masking (DCM) (Davies et al., 2023; Prakash et al., 2024).

These methods provide a time-efficient way to search for human-interpretable variables encoded in intractably large or innumerable mediator sets. However, current optimization-based methods can only search for causal nodes given a clear pre-existing causal hypothesis of how a model accomplishes some behavior, and/or labeled data. These methods can be evaluated with respect to accuracy in capturing model behavior, but they do not directly indicate *a priori* what those hypotheses should be, or in what specific ways our hypotheses are wrong. They also require sufficient training data to demonstrate the behavior of interest. As with all parametric methods, the above approaches are subject to overfitting or underfitting, which can be a key concern when not enough data is available.

### 5.2.2 Unsupervised mediator searches

Supervised search methods (see §5.2.1) require specific hypotheses about the internal representations of neural networks. However, neural networks implement various behaviors, many of which may be counterintuitive to humans and therefore more likely to be missed. For example, while Li et al. (2023) hypothesized a constant board state representation in Othello, Nanda et al. (2023) later found that the model actually switches the board state representation with every turn, taking the view of "my pieces vs. opponent's pieces" rather than "black pieces vs. white pieces". Therefore, it can be desirable to use techniques for searching for mediators without specifying a hypothesis ahead of time.

Hence, some studies employ *unsupervised* methods. Because these techniques are unsupervised, they return a large—but finite—collection of mediators. Unsupervised methods are largely *correlative*, meaning that the discovered mediators may not necessarily capture causally relevant or faithful aspects of the NN's computation. However, the discovered mediators can then be implicated in NN computation post-hoc by employing additional techniques, such as those from §5.1, to select task-relevant mediators from this collection.

**Feature disentanglement using sparse autoencoders.** Exhaustive search for meaningful non-basis-aligned directions is impossible due to the infinite search space. The *feature disentanglement* literature tackles this problem by performing an unsupervised search for directions in neuron activations which both (1) capture the information encoded in the internal representations and (2) are *disentangled* from other meaningful directions. Bengio et al. (2013) characterize disentangled representations as *factors for variations* in the training dataset. To identify these factors of variations, Sharkey et al. (2023) used sparse autoencoders (SAEs) to perform dictionary learning on a one-layer transformer, identifying a large (overcomplete) basis of features. SAEs are trained to reconstruct the input activations while only activating a sparse subset of dictionary features. Cunningham et al. (2024) then applied SAEs to language models and demonstrated that the observed dictionary features are highly interpretable and can be used to localize and edit model behavior. Since then, numerous researchers have explored this area (Templeton et al., 2024; Rajamanoharan et al., 2024; Braun et al., 2024; Bricken et al., 2023, *inter alia*). In practice, SAEs have shown initial promising results in identifying functionally relevant and human-interpretable features. However, they are not able to perfectly reconstruct the internal activations. Most importantly, however, we do not know *a priori* what the ground truth features are in the model's computation, and can only use the reconstruction performance as a proxy measure of performance.

**Correlation-based clustering.** Another unsupervised way of discovering meaningful units is clustering mediators by the similarity of their behavior. This idea is not new (cf. Elman, 1990), but running causal verifications of the qualitative insights from clustering studies is relatively rare. Dalvi et al. (2020) cluster neurons, and are able to maintain performance when ablating a significant portion of them; the goal of this study was not interpretability, but their results nonetheless causally verify that redundancy is very common in neural networks.

There has recently been renewed interest in clustering-based mediator search. Michaud et al. (2023) propose a method to identify interpretable behaviors within neural networks by clustering parameters. Because the identified behaviors tend to be coherent, the units implicated in each cluster can be viewed as a set of components that have a functionally coherent role in the network. Marks et al. (2024) and Engels et al. (2024) generalize this from gradients to neuron or sparse autoencoder activations. The activations that compose the clusters are then labeled according to the dataset samples on which they activate most highly. These clusters are a subset of all mediators which are relevant to performing some prediction task; thus, one could perform interventions to the mediator sets that compose a cluster. This idea has not yet been extensively employed or explored. However, ablating the elements within these clusters could be a useful way to establish the functional role of *groups of* components in future work, or assess whether a subset of a model's behavior is implicated in a more complex task. We discuss this in §7.

# 6 Related Work

Causally-grounded interpretability surveys do not always focus on model internals, and surveys that focus on model internals do not necessarily require causal grounding. We give a brief overview of both types here. We also discuss recent tooling efforts that have accompanied and accelerated the growing interest in mechanistic and causal interpretability.

**Mechanistic/model-internal interpretability surveys.** Some surveys catalogue studies that aim to understand the latent representations of neural networks (Belinkov & Glass, 2019; Danilevsky et al., 2020; Belinkov, 2021; Sajjad et al., 2022); these have often called for more causal validations of correlational observations. More recent surveys tend to focus increasingly on categorizing or giving overviews of methods for intervening on model internals (Ferrando et al., 2024; Rai et al., 2024), understanding the trajectory of the mechanistic interpretability field (Räuker et al., 2023), and/or cataloguing the impacts of the field (Bereska & Gavves, 2024). We propose a more theoretically grounded framing, and categorize interpretability work from many domains as part of the causal interpretability literature. Moreover, we treat the units of causal analysis that a study employs, as well as the way in which the study searches over those units, as primary factors in categorizing the study; this is in contrast to the above-cited surveys, which categorize studies according to the class of method rather than the unit of analysis.

**Causal interpretability surveys.** Moraffah et al. (2020) is a causal interpretability survey that categorizes various streams of causal interpretability research according to the methods they employ, though the catalogued studies are not necessarily based in the ideas of causal mediation analysis nor aimed toward understanding model internals. Other interpretability surveys (Subhash et al., 2022; Gilpin et al., 2018; Singh et al., 2024a) focus on methods for explaining the decisions of neural networks without prioritizing causally grounding the explanation methods or focusing on model internals. Many causality-focused surveys are domain-specific, including areas such as cybersecurity (Rawal et al., 2024) and healthcare (Wu et al., 2024b). Some focus on particular domains; for example, in NLP, some focus on how causal inference can improve interpretability (Feder et al., 2022), or ways to explain (Lyu et al., 2024) or interpret (Madsen et al., 2022) neural NLP systems.

**Tools.** Several libraries have recently been released to facilitate causal interpretability methods that involve interventions to model components. These tools can implicitly prioritize certain types of mediators over others. For instance, `pyvene` (Wu et al., 2024d) is designed specifically to aid in locating non-basis-aligned multidimensional subspaces via alignment search methods such as DAS and its successors (Geiger et al., 2024; Wu et al., 2023; Huang et al., 2024; Wu et al., 2024c). While it can also be used for other kinds of model interventions, this library could be particularly useful for those wishing to verify existing causal hypotheses. `TransformerLens` (Nanda & Bloom, 2022) and libraries based on it (`Prisma`; Joseph, 2023) are interpretability tools for examining Transformer-based neural networks. Its standardized interface across model architectures tends to encourage a focus on basis-aligned components such as neurons and attention heads, subspaces, and layers, as interventions to these mediators are natively supported. `NeuroX` (Dalvi et al., 2023) similarly incentivizes neuron-level interpretability in particular. `NNsight` (Fiotto-Kaufman et al., 2024) and `Baukit` (Bau, 2022) are more transparent interfaces that provide access to the underlying PyTorch model architecture, which allows for flexible modifications of the model's computation graph. Due to different naming conventions of model developers, this more transparent access may make it harder to generalize basis-aligned intervention code across architectures at first, but research on both basis-aligned and non-basis-aligned mediators is more accessible under this paradigm.

We have mainly discussed these toolkits with respect to the mediators that they enable working with. Note that this survey is intended more as a new perspective rather than a practical guide to using these tools. Some surveys such as as Ferrando et al. (2024); Rai et al. (2024) or code tutorials (Nanda & Bloom, 2022; Wu et al., 2024d; Fiotto-Kaufman et al., 2024) make hands-on practical introduction to particular methods their explicit purpose, without necessarily assuming (nor discussing the benefits nor drawbacks of) a particular theory; see these for more hands-on guides to implementing interpretability methods.

# 7 Discussion and Conclusions

## 7.1 What is the right mediator?

There are pros and cons to any mediator, and the best mediator will therefore depend on one's goals. In this section, we ask: When is it appropriate to deploy particular kinds of mediators and search methods? We then give concrete examples of past works that have deployed these mediator/method combinations to accomplish particular goals.

**Explaining model behavior.** If the goal is to explain model behaviors, then in the absence of compute restrictions and with no strong prior hypotheses as to how a model performs some behavior, **unsupervised optimization-based methods over fine-grained mediators** (such as non-basis-aligned directions) provide a strong starting point. For example, unsupervised methods like sparse autoencoders provide a fine-grained and human-interpretable interface to a model's computation. That said, autoencoder features are not guaranteed to be faithful to a neural network's behavior in next-token prediction, and they require either a human or an LLM to label or interpret the features, which is laborious and expensive. Moreover, natural language explanations of neurons and features have inherent flaws (Huang et al., 2023): they may often exhibit both low precision and recall. Second, interpretable mediators like directions require more human effort and/or compute than basis-aligned components to locate, and one may need to rediscover these directions if fine-tuning, adapting, or editing is part of the study.[16]

**Verifying a mechanistic hypothesis.** If one has a clear idea as to how a model accomplishes a task and simply wishes to verify a mechanistic hypothesis, then **multidimensional non-basis-aligned subspaces** may be the right mediators, and a reasonable corresponding search method would be **counterfactual-based optimization**. One can automatically search for the subspaces which correspond to a particular node in one's hypothesized causal graph using alignment search methods, as in Geiger et al. (2024); Wu et al. (2023). Alignment search entails learning a rotation to some activation subspace given an input, and then performing causal interventions by substituting activations from other runs in the rotated latent space. This allows us to locate distributed representations that act as single causal variables in non-basis-aligned spaces. This is relatively scalable, and enables us to qualitatively understand intermediate model computations. The primary downside is that we must anticipate the mechanisms that models employ to perform a task; if we cannot anticipate them, then curating data and refining one's causal hypotheses may require significant human effort. Additionally, the same causal graph could correspond to many different qualitative explanations, depending on the data used to discover the causal graph. Finally, this mediator type is subject to the same confounds as other optimization-based techniques: the optimization itself could learn the phenomenon, rather than extracting it from the model of interest.

**Localization and editing.** If one's goal is to localize some phenomenon in a model, and not to understand *how* a model implements a behavior *per se*, then **exhaustive searches over basis-aligned subspaces or full layers** may be sufficient. There are many comprehensive causal techniques for locating these, including causal tracing (Meng et al., 2022) and activation patching (Vig et al., 2020), as well as techniques for locating graphs of basis-aligned mediators, such as circuit discovery algorithms (Goldowsky-Dill et al., 2023; Wang et al., 2023; Conmy et al., 2023). Some of these methods are relatively slow in their exact form, but fast approximations exist to these causal metrics, including attribution patching (Syed et al., 2023) and improved versions thereof (Kramár et al., 2024; Hanna et al., 2024; Marks et al., 2024). Even in the absence of a deep understanding of the role of these mediators, localization can be useful for downstream applications like model editing (Meng et al., 2022; 2023)[17] and model steering (Todd et al., 2024; Goyal et al., 2020). That said, if meaningful features are not actually aligned with neurons/heads, then we are not guaranteed to get the best performance until we move beyond basis-aligned spaces. Future work should analyze the performance of model editing and steering methods when using different kinds of mediators. For example, Marks et al. (2024) compare the efficacy of ablating neurons versus sparse features (non-basis-aligned directions), and

---

[16]Though Prakash et al. (2024) find that the same model components are implicated in an entity tracking task before and after fine-tuning.

[17]Though Hase et al. (2023) find that causal localizations do not always reflect the optimal locations for editing models.

find that ablating sparse features is significantly more effective; the difference may be much smaller at the coarse granularity of full layers and submodules, but there is not yet much empirical evidence on what kinds of mediators are most effective for particular applications.

**Examples.** Assume you are a researcher interested in understanding how a language model performs multiple-choice question answering. Your lab can afford to rerun fine-tuning and adaptation if this model is particularly bad, so you do not intend to perform precise model editing based on the results of your evaluations or interpretability experiments. Instead, you care mainly about predicting success and failure modes on future examples so that you know whether this model could be deployed in production, and in what cases you should double-check the model's outputs. In this case, your goal is to explain model behavior, and you do not have a specific mechanistic hypothesis. Thus, you should deploy an unsupervised method to locate and search over meaningful features. This will help you find unantipicated mechanisms.

Assume instead that you want to precisely edit the knowledge of the model on cases where it gets the answer wrong. Now, you do not care as much about interpreting the model's general answering process, but rather, debugging and fixing specific mistakes. Thus, this would fall under localization and editing: you want to locate the source of incorrect answers, and patch them to improve performance. Thus, as a first step, you should deploy an exhaustive search over a relatively coarse-grained mediator, such as submodules or full layers. You can use model editing techniques like ROME or MEMIT to edit facts in cases where the model answered incorrectly. It is theoretically possible that localizing facts and editing over finer-grained mediators could result in even better performance, but this has not yet been attempted. Future work should investigate whether this is possible, and whether the expected time complexity increase is worth the performance improvements.

Now assume that you are running a different kind of study: your task is still multiple-choice question answering, but you are testing a specific hypothesis as to how the model accomplishes the task. You want to know whether it represents "questionhood" separately from factual statements, and your study is only concerned with to what extent this holds—not accuracy on the task *per se*. Here, you want to verify a specific mechanistic hypothesis, so you should design a dataset of counterfactual pairs that isolates this variable, and then deploy counterfactual-based optimization over multidimensional non-basis-aligned subspaces. This will yield a set of scores that indicate to what extent the hypothesis causally explains the model's output behavior.

Note that none of these examples have recommended the use of basis-aligned subspaces, such as (sets of) neurons or attention heads. This is not to say that they are not useful, but it does indicate that when compute is not a significant limitation, they are often not the best place to start when working with realistic neural networks trained on large-scale data. These units are often difficult to interpret, and there are many of them; other mediator types resolve at least one of these issues. That said, basis-aligned subspaces may be useful when we expect that they may have interpretable meanings (e.g., in toy task settings), or when we expect that unsupervised methods like sparse autoencoders are likely to yield bad results or are simply not effectively trainable.

### 7.2 Suggestions for Future Work

By centering the unit of causal analysis rather than the method, we can gain new insights into the kinds of research that will be necessary to advance causal interpretability. Here, we discuss lines of work we believe will be fruitful.

#### 7.2.1 Are there better causal mediators?

There are almost certainly better causal mediators that have not yet been discovered. Current work on improving mediators tends to focus on non-basis-aligned directions, such as sparse features or directions discovered from supervised probing on the activations of a single layer/submodule. One could consider pursuing coarser-grained mediators by discovering multi-layer **model regions** or **component sets** which accomplish a single behavior. Because these regions can cross layers, they would include non-linearities that allow them to represent more complex functions or concepts.

**Non-linear and multidimensional feature discovery.**   As discussed in §4.4, there is recent work demonstrating the existence of human-interpretable *multidimensional* features. For example, days of the week are encoded circularly as a set of 7 directions in a two-dimensional subspace (Engels et al., 2024), and current methods cannot easily capture such multidimensional features. Group sparse autoencoders (Theodosis & Ba, 2023) or clusters of autoencoder features could be a way to capture multidimensional non-basis-aligned features in an unsupervised manner, but thus far, empirical work has not yet demonstrated whether this will be effective for interpreting neural networks. Additionally, many current causal interpretability methods require binary distinctions between correct and incorrect answers, whereas causal mediation analysis does not have any theoretical linearity, dimensionality, or Boolean restrictions.

There may also exist higher-order non-linear concepts in latent space that we have not yet been able to find, due to the linear focus of contemporary methods. For example, a subgraph or subcircuit can encode a coherent variable representation or functional role, as in Lepori et al. (2023). How can we discover these subgraphs? Path patching (Goldowsky-Dill et al., 2023; Wang et al., 2023) provides a manual approach to implicating subgraphs as causal mediators, we do not yet have automatic methods that can scalably search over all possible subgraphs in a network. How might non-linear and/or coarse-grained mediators like these be useful in practice? As an example, we would expect fundamental phenomena like syntax to be implicated in downstream tasks like question answering, if we expect that language models are robustly parsing the meaning of their inputs. Thus, we could implicate the entire syntax region(s) in the model's final decisions; if it is not strongly implicated in QA performance, then we have a strong hint that the model may instead be relying on a mixture of surface-level spurious heuristics to parse inputs.

### 7.2.2   Inherently interpretable model components

More ambitiously, one could consider building models with inherently interpretable components—i.e., whose fundamental units of computation (or some subset thereof) are designed to be sparse, monosemantic, and/or human-interpretable, but ideally still expressive enough to attain good performance on downstream tasks. Examples based in neural networks include differentiable masks (De Cao et al., 2020; Bastings et al., 2019), transcoders (Dunefsky et al., 2024), codebook features (Tamkin et al., 2023), and softmax linear units (Elhage et al., 2022a). Many of these are primarily post-hoc components that decompose model components into interpretable units, but they could potentially be integrated into the network itself during pre-training alongside a loss term (in addition to the language modeling loss) that enables fine-grained interpretability at all stages of pre-training.

Alternatively, more focus could be devoted to building models that are designed from the ground up to be interpretable, such as backpack language models (Hewitt et al., 2023) and concept bottleneck models (Koh et al., 2020; Oikarinen et al., 2023); a related idea is to train the model using loss terms that encourage success on intermediate tasks, or induce particular kinds of feature representations (Hupkes et al., 2017). Perhaps least invasively, we could consider pre-training methods that softly encourage interpretable features to be aligned to neuron bases; this would remove the need for optimization to find non-basis-aligned components, and therefore make interpreting NN decisions significantly easier and less confounded. However, this would reduce the number of features that could be encoded per neuron, so it would likely require significantly more parameters, or accepting degradations in performance.

### 7.2.3   Scalable search

As the size of neural networks increases, the number of potential mediators to search over will also increase. The situation worsens as we start searching over continuous sets of fine-grained mediators such as non-basis-aligned directions. Although a few gradient-based or optimization-based approximations to causal influence have been proposed to improve time efficiency, such as attribution patching (Syed et al., 2023) and DCM (Davies et al., 2023), more work is still needed to evaluate the efficacy of these techniques in identifying the correct causal mediators. Additionally, better techniques beyond greedy search methods should be devised to identify causally important *groups of* mediators; these should aim to produce Pareto improvements over time complexity and causal efficacy.

As discussed in §5.2.1, optimization-based mediator search methods often require a pre-existing hypothesis about how a model implements a particular behavior of interest. Another path toward scaling mediator search and interpretability is to automate the process of hypothesis generation. Qiu et al. (2024) showed that current LLMs can generate hypotheses, and Shaham et al. (2024) showed that hypothesis refinement via LLMs can aid humans in interpreting the causal role of neurons in multimodal models. Similary, LLMs could be used to automate and scale hypothesis generation regarding the role of particular mediators across a wider variety of tasks and models. Optimization-based based methods such as DAS or DCM could then be used to causally verify the automatically generated hypotheses.

### 7.2.4 Benchmarking progress in mechanistic interpretability

Another key direction will be **standard benchmarks for measuring progress in mechanistic interpretability**. Currently, most studies develop ad-hoc evaluations, and generally only compare to similar methods that employ the same mediators. Thus, to measure whether new mediators or search methods are truly giving us improvements over previous ones, we need to develop principled methods for direct comparisons. In circuit discovery, it is theoretically possible to use the same metrics to compare any circuit discovered for a particular model and task, regardless of whether sparse autoencoders are used, whether the circuit is based on nodes or edges, among other variations. Direct comparisons like these are not standard, though some recent work has begun to perform direct comparisons across mediator types, such as Miller et al. (2024). Huang et al. (2024) propose to directly evaluate interpretability methods according to the generality of the abstractions they recover, and do directly compare across different mediator types given the same model and task. Arora et al. (2024) and Makelov et al. (2024) also propose standardized interpretability benchmarks that allow us to compare across mediator search methods, though they do not directly compare across mediator types.

Direct comparisons require defining criteria for success, but there is little agreement about the kinds of phenomena we should be measuring, and precisely how they should be measured. Taking circuit discovery as a case study, there are (at least) five key metrics: (1) **faithfulness**, or how well the circuit captures the full model's behavior; (2) **generalization**, or how well the circuit generalizes outside the distribution of the dataset used to discover it; (3) **completeness**, or whether we have captured all relevant components; (4) **minimality**, or whether we have not included any superfluous components; and (5) **interpretability**, which refers to how human-understandable the circuit is. Some of these metrics currently have multiple contradictory senses: (1) is sometimes used to refer to whether the circuit captures the full model's behavior (including productive *and* counterproductive components), and in other cases is used to refer to whether the circuit only captures components that result in high performance on the task.[18] The choice of mediator will also significantly affect how well we can perform each of these, as obtaining a complete and faithful circuit requires working at the correct level of abstraction (i.e., where meaningful units of computation are actually represented). Additionally, these metrics can be difficult to automatically measure; for example, (5) may require human evaluations at first (Saphra et al., 2024), as many NLP tasks such as machine translation required before the introduction of now-standard[19] metrics such as BLEU, COMET, and METEOR scores (Papineni et al., 2002; Rei et al., 2020; Banerjee & Lavie, 2005). Some tasks have started to integrate human/user evaluations, which will be especially useful for building interpretability tools that are grounded in real-world use cases and settings (Saphra et al., 2024). Future work could consider defining a broad set of models, target tasks, and informative metrics on which researchers can compare their interpretability methods, mediator types, and search methods in a standardized way. This will enable us to assess whether new search methods and mediator types are producing real Pareto advancements with respect to interpretability, efficiency, and description length.

For many methods and tasks, more task-specific measures and benchmark datasets will be needed. For example, Cohen et al. (2024) and Zhong et al. (2023) propose benchmarks to evaluate model editing methods on out-of-distribution examples, and Karvonen et al. (2024) propose to measure progress in feature disen-

---

[18]Even if we decide on one of these definitions, there is still the issue of how to compare the full model to the discovered subgraph. We could use KL divergences, compare probabilities of a given token, among other possibilities; Zhang & Nanda (2024) investigate this in detail.

[19]"Standard" does not necessarily mean "representative of human judgments".

tanglement using board game models. While not the main focus of this paper, we believe that building standardized benchmarks will be a key means to the end of assessing whether advancements in causal mediators are producing real gains in interpretability. More robust evaluation metrics and methods will lead to better science, which will ideally allow us to assess whether new causal abstractions are fundamentally more useful—both for describing the computations of a neural network *and* for practical applications.

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

# A    Types of interventions

Recall that to compute the indirect effect, we must replace the activation $z$ of a mediator $Z$ with some counterfactual activation $z'$. In effect, the goal of an intervention is to reveal a neuron for whom swapping $z$ with $z'$ will produce a large effect on the output metric. There are many ways to derive $z'$; some of these depend on $x$, some depend on the expected output, and some depend on neither. We describe each of these classes of interventions, and list their pros and cons.

**Input-dependent interventions.**   If one cares about neurons that are sensitive to a specific contrast, then one can use input-dependent interventions (i.e., interventions where $z'$ depends on $x$). For example, assume our target task is subject-verb agreement. Given an input $x = $ "The **key**", we want to locate neurons that increase the probability difference $m = p(\textbf{is}) - p(\textbf{are})$. In this case, we can obtain $z$ by running $x$ through model $M$ in a forward pass (denoted $M(x)$) and storing the activation of component(s) $a_i$ (which could be a neuron, for example). We can then obtain $z'$ by running $M(x')$, where $x'$ is a minimally different input that swaps the answer; here, $x'$ would be "The **keys**". This type of intervention preserves all example-specific information, and varying only the grammatical number of the subject. This makes this intervention precise: it will only reveal neurons for which swapping grammatical number *and nothing else* will significantly affect $m$.

This method is easily controllable and reveals components targeted to a specific phenomenon; in other words, this is a high-precision intervention. However, humans must carefully curate highly controlled input pairs in which only one phenomenon is varied across $x$ and $x'$. Additionally, input-dependent interventions work best for (and arguably, are only principled given) binary contrasts: we contrast two minimally different inputs, which isolates neurons sensitive only to the difference between items in the pair. When working with categorical or ordinal variables, it is not immediately clear how to construct $x'$ to recover all relevant components. Additionally, it does not recover all task-relevant components, but rather those related to the contrast between $x$ and $x'$; in other words, this is a low-recall method. For instance, Vig et al. (2020) showcases the requirement of enumerating through all possible gender pronouns and personas related to a specific gender to measure the specific phenomenon, even noting that full generalizability to all grammatical gender pronouns is difficult. Furthermore, such interventions can be privy to unreliable explanations as shown in Srivastava et al. (2023) wherein the input data can be corrupted to manipulate the concept assigned to a neuron. Hence, input-dependent interventions may require additional safeguards to ensure safety and fairness in critical real-life applications.

**Class-dependent interventions.**   In contrast to input-dependent interventions, which locate components sensitive to a specific phenomenon, class-dependent interventions define a single intervention to produce some behavior associated with a class of outputs. For example, Li et al. (2024) learn an ablation mask over the computational graph of a language model to prevent the model from producing toxic content; here, the classes are *toxic* and *not toxic*, and the intervention is the same regardless of the input.

Class-dependent interventions provide a single flexible intervention that works for any given input. However, they require a dataset of input-label pairs that can be used to learn the interventions. This is often time-consuming and can lead to errors as many labels we care about are hard to quantify (e.g., bias or toxicity).

**Class- and input-independent interventions.**   This type of intervention does not rely on either the input or a class label, and its goal is generally to fully remove the information encoded by a mediator—regardless of whether the information is task-relevant. A common ablation type is **zero ablations**, where the activation of a component is set to 0. This is not entirely principled, since 0 has no inherent meaning in an activation—for example, a neuron's default activation may be non-zero, whereas 0 itself is out of distribution relative to what the model expects. A more principled ablation type is a **mean ablation**, where the neuron's activation is set to its mean value over some distribution—either task-specific data or general text data. A **resampling ablation** is a special case of mean ablations where the sample size is 1.

This is a more general intervention type that can be run without access to contrastive input/output pairs, and without labeled inputs. It allows us to tell whether *any* of the information in a mediator is necessary for a model to perform the task, but it may also affect other information in unanticipated ways; in other words,

it has high recall and low precision relative to other intervention types. It also may cause performance on a task to drop in a way that reveals spurious mediators, rather than mediators that are conceptually relevant—for example, ablating a neuron that detects the word "dog" may reduce the probability of the correct verb form "is" over the incorrect verb form "are", but this is a highly specific neuron that does not on its own reveal general information about how models process subject-verb agreement.

