# OpenReview forum: "The Quest for the Right Mediator: A History, Survey, and Theoretical Grounding of Causal Interpretability"
_TMLR — Rejected by TMLR_

### Review · Reviewer_J7Fd · 2024-10-09

**Summary Of Contributions:**

- Causal interpretation of the impact of input features on output results.
- Importance of mediation analysis in the context of neural networks.
- History of the evolution of causal explanation in recent times.
- Current status of research methods for mediators in neural networks.
- Other surveys, tools, benchmarks.

**Audience:**

Yes

**Broader Impact Concerns:**

None.

**Claims And Evidence:**

Yes

**Requested Changes:**

Critical issues:

- The material must be connected to the causal graph framework in a less vague way: What are causes and effects in the vast neural networks landscape? How do we compute the impact of a mediator (whatever computation unit we choose) on the outcome? How can we assess the relevance of the mediators?
- While the material is structured in sections, subsections and paragraph, authors do not provide an easy and compact way to navigate it. A survey is effective if it provides an effective and informative way of navigating the hundreds of references cited in it, rather than a list of papers. Consider a schema, table, graph-like support to understand why a given contribution does or not qualify as relevant w.r.t. the task of selecting a mediator.

Major issues:

- The related work section can be improved significantly: Why should I read this long survey rather than another one? If I want to start exploring this field now, how the cited tools can be used? Same remarks to Subsection 7.2.4 about benchmarking.

Minor issues:

- Consider adding modules/submodules elements to Figure 2.

**Strengths And Weaknesses:**

Strengths:

- The survey encompasses a vast chunk of literature on neural network explainability.
- The survey provides a comprehensive review of the history of the available material from an alternative point of view.

Weaknesses:

- Section 2 - Preliminaries are virtually non-existent: the definitions of the causal quantities are approximate, vague and refer to a small chunk of the overall literature about causality. What is the definition of a causal graph? What is a cause? What is an effect? Why knowing the definitions of such theoretical objects is relevant in the context of this survey? Mediators are known also for other counterfactual estimates, e.g. natural effects. If the core of the paper is about choosing the "right mediator", what would be the right decision criterion?
- Section 3/4/5 are loosely connected to the causal graph framework. Most of the cited works seems to look for a specific computational unit with some sort of causal interpretation linked to the input features. However, the general message is that no unit is capable of representing an isolated computational subgraph that is sensitive to a given input feature. How does this relate to the semantic specific interpretation provided by causal graphs? Is the search of the right mediator feasible or it is just a vague analogy to interpretation? How the cited papers are actually mapped in some way to Pearl's theory? What about the identification phase of the right causal estimand?
- Section 5 - Tools are invaluable starting points to inexpert readers, this part of the survey is currently limited and could be expanded significantly, making this contribution more actionable.

---

> ### Author Response · Authors · 2025-01-10
> **Response to J7Fd**
>
> Thank you for the insightful and theoretically grounded comments.
>
> > [T]he definitions of the causal quantities are approximate, vague and refer to a small chunk of the overall literature about causality. What is the definition of a causal graph? What is a cause? What is an effect? Why knowing the definitions of such theoretical objects is relevant in the context of this survey?
>
> Our perspective crucially relies on viewing neural networks as causal graphs. Our analysis is concerned primarily with the **nodes** in causal graphs rather than the graphs composed thereof, but we agree that defining and discussing causal graphs would give helpful context to readers, and help them better understand why we care about these nodes. We had a definition of causal graphs on page 2 under the paragraph titled “Causal abstractions in neural network interpretability”, which we have expanded significantly. See highlighted changes in Section 2.
>
> We believe that the causal graph framework is a valuable abstraction that allows us to explain all of these concepts in a concise way. It is also a helpful way to think about neural networks, as their computation graphs are already graphical structures! Thus, the entire focus of interpretability can be viewed as the process of transforming non-human-understandable computation graphs into more human-interpretable and abstract causal graphs.
>
> >  If the core of the paper is about choosing the "right mediator", what would be the right decision criterion?
>
> Section 7.1 is dedicated to this topic. In this section, we ask: given a particular research goal, what kind of mediators are best? To summarize, we state that fine-grained mediators, such as autoencoder features or neuron-level mediators, are best if one’s goal is to understand how a model accomplishes a task. If one’s goal is merely to figure out which parameters to prune or modify to improve performance, more coarse-grained mediators such as layers and submodules are acceptable. Time permitting, we would appreciate elaboration on this point!
>
> > Most of the cited works seems to look for a specific computational unit with some sort of causal interpretation linked to the input features. However, the general message is that no unit is capable of representing an isolated computational subgraph that is sensitive to a given input feature. How does this relate to the semantic specific interpretation provided by causal graphs? […] How the cited papers are actually mapped in some way to Pearl's theory?
>
> We believe there is a subtle and reasonable misunderstanding here: we do not claim that a specific single mediator can ever capture the full subgraph encoding a particular model behavior. Rather, we are concerned with the type of units that the causal graphs are composed of. Should the nodes in our causal graphs be neurons, or attention heads, or something more abstract like linear combinations of these units (Sec. 4)? If a particular type of unit is better than others, why (Sec. 4, Sec. 7.1)? And once we’ve decided, how do we find the specific units of those types that are most impactful for a given task (Sec. 5)?
>
> The mapping to Lewis’ and Pearl’s theories lies in the fact that we are always measuring the causal influence of a mediator on a target behavior using the **indirect effect**, which encodes a continuous notion of counterfactual influence. This requires us to implicitly construct a structural model whose structure is derived from the computation graph. We illustrate the causal framework in Figure 2. Thank you for this feedback!
>
> > Is the search of the right mediator feasible or it is just a vague analogy to interpretation?
>
> We understood this question to mean "Is there ever a case where there is a truly ‘right’ mediator? Or does it depend on the level of interpretation that one wants?” This is a great question, and one we address in detail in Section 7.1. See discussion above.
>
> > What about the identification phase of the right causal estimand?
>
> See response to comment above about choosing the right mediator, and the discussion below about the indirect effect metric.
>
> > Tools are invaluable starting points to inexpert readers, this part of the survey is currently limited and could be expanded significantly, making this contribution more actionable.
>
> We have expanded this section to include more discussion on actionable takeaways, and explain why we did not focus more extensively on this throughout the paper.
>
> > The material must be connected to the causal graph framework in a less vague way: What are causes and effects in the vast neural networks landscape?
>
> See response to above comment on the definition of causal graph, node, edge, cause, and effect.

---

> ### Author Response · Authors · 2025-01-10
> **Response to J7Fd (Pt. 2) and References**
>
> > How do we compute the impact of a mediator (whatever computation unit we choose) on the outcome? How can we assess the relevance of the mediators?
>
> We define and discuss a commonly used metric for this purpose in Section 2: the **indirect effect**. This is a causal metric that measures the counterfactual influence of a cause on an effect. Generally, when assessing the relevance of a mediator, we compute the indirect effect of the mediator on a particular model behavior. We have made the connection between Lewis’ and Pearl’s counterfactual theory and this metric clearer.
>
> > While the material is structured in sections, subsections and paragraph, authors do not provide an easy and compact way to navigate it.
>
> Thank you, this point was also raised by Reviewer EVLS. We have added a summary figure to the end of the Introduction, a Table that summarizes the mediator types and search methods, more cross-references to improve navigability.
>
> > Why should I read this long survey rather than another one? If I want to start exploring this field now, how the cited tools can be used?
>
> We argue that the perspective introduced by this survey encourages us to think more about the pros and cons of particular mediator types. That is: when investigating why a neural networks behaves in a particular way, or which components contribute to a given behavior, what do we gain and lose by basing this investigation on neurons, or attention heads, or more abstract components like autoencoder features? This makes it easier to know when particular units of analysis should be used, given one’s research goals. On a more theoretical level, it also gets at a fundamental question: What are the right terms and abstractions for discussing the behaviors of a neural model, such as a language model? This is an important open question that much mechanistic interpretability work is currently trying to address.
>
> We view the survey as more of a perspective, rather than a practical guide. Papers such as [1,2] make hands-on practical introduction to particular methods their explicit purpose, without necessarily assuming (nor discussing the benefits nor drawbacks of) a particular theory.
>
> > Consider adding modules/submodules elements to Figure 2.
>
> We have added submodules to this figure. Thank you!
>
> References
> ===
> [1] Rai et al. (2024). “A practical review of mechanistic interpretability for transformer-based language models.” https://arxiv.org/abs/2407.02646v1
>
> [2] Ferrando et al. (2024). “A primer on the inner workings of transformer-based language models.” https://arxiv.org/abs/2405.00208v1

---

### Review · Reviewer_Eyeu · 2024-10-29

**Summary Of Contributions:**

The focus of the present work is on causal interpretability through causal mediation analysis. Firstly, notions like "counterfactual interventions", "mediators" and "neurons" are introduced and their use on the topic is described. Throughout the paper the authors narrate the history of causal interpretability discuss the pros and cons of several methods that have been used up to date. They then argue that the employed mediators should be chosen based on the goals of each study. Lastly, they also flag the need for employing standard evaluation methods in order to be able to perform principe comparison across all mediator types.

**Audience:**

Yes

**Broader Impact Concerns:**

no concerns

**Claims And Evidence:**

Yes

**Requested Changes:**

It would be appreciated if the authors could please:
1) revise the notation so that is consistent throughout the paper;
2) revise the terminology and abbreviations/acronyms, e.g. MLP abbreviation is employed from page 1 but the"multi-layer perceptrons" term is used in page 3, without linking the abbreviation, also NLP is mentioned in page 13 but the acronym is not explained;
3) the flow of the paper is somewhat confusing as, e.g., the term "neuron" is core since page 2 but it is defined/explained in page 7

**Strengths And Weaknesses:**

Strengths:
-- thorough narration of the course of causal interpretation from Lewis 1973,  to 2024 state of the art approaches


Weaknesses:
-- abuse of notation (page 6, not consistent wrt neuron and activations notation, variable "m" represents more than one concepts)
-- lack of contribution: albeit a thorough survey on the history of causal interpretation, this paper lacks any novelty or insights

---

> ### Author Response · Authors · 2025-01-10
> **Response to Eyeu**
>
> >  [A]buse of notation (page 6, not consistent wrt neuron and activations notation, variable "m" represents more than one concepts)
>
> Thank you for this feedback; we have reconsidered our notation. We now use capital letters to represent random variables, and lowercase letters to represent specific values that these variables can take. When discussing neural networks, we use bold letters to represent vectors, bold letters with subscript indices to represent elements of these vectors, and normal math-font letters to represent specific scalar values that these elements can take. To summarize:
> * $m$ refers to the target metric, which is a scalar. This could be defined in many ways depending on the study, but usually refers to a token probability or logit (difference).
> * $X$ and $Y$ refer to the input and output variables of a causal graph, respectively. $x$ is a specific input, and $y$ is a specific output.
> * $Z$ is a generic variable referring to any mediator (intermediate node) between $X$ and $Y$. $z$ and $z’$ are natural and counterfactual values, respectively, that $Z$ can take given $X=x$.
> * $\ell$ is a layer index (e.g., layer 2).
> * $\mathbf{a}^\ell$ is an activation vector at the output of layer $\ell$. $\mathbf{a}^\ell_i$, a dimension of $\mathbf{a}^\ell$, is a neuron in this vector. $a_i^\ell$ is an activation of neuron $\mathbf{a}_i^\ell$.
>
> > Lack of contribution: albeit a thorough survey on the history of causal interpretation, this paper lacks any novelty or insights
>
> We claim that the perspective introduced by this survey encourages us to think more about the pros and cons of particular mediator types. That is: when investigating why a neural networks behaves in a particular way, or which components contribute to a given behavior, what do we gain and lose by basing this investigation on neurons, or attention heads, or more abstract components like autoencoder features? This makes it easier to know when particular units of analysis should be used, given one’s research goals. On a more theoretical level, it also gets at a fundamental question: What are the right terms and abstractions for discussing the behaviors of a neural model, such as a language model? This is an important open question that much mechanistic interpretability work is currently trying to address.
>
> In the paper, we state these contributions, and give examples of the ways in which this perspective enables us to more easily see new actionable directions. For example, much of the recent sparse autoencoder trend can be attributed to the fact that these allow us to efficiently and easily interpret combinations of activations from multiple neurons (Sec. 4). Innovating on mediator types can thus lead to significant gains in interpretability and model control. It also makes clear when using certain kinds of  mediators makes the most sense for particular kinds of studies; see Sec. 7.1. More generally, we also point out open directions for devising even better mediators (Sec. 7.2.1), and discussing combinations of mediators and search methods that have not yet been tried (Sec. 7.2.3).
>
> Time permitting, if you believe these are insufficient contributions, could you elaborate more specifically? Does this comment refer more to a perceived lack of novelty in this perspective, or more to a perceived lack of actionable insights given this perspective?
>
> >  [R]evise the notation so that is consistent throughout the paper.
>
> We have made changes to the notation. These are highlighted in the revised PDF. Please let us know if you have any remaining concerns!
>
> > Revise the terminology and abbreviations/acronyms.
>
> Thank you. We have addressed the MLP/NLP concerns, and have combed through the paper for other instances of abbreviations not being presented at the first instance of a phrase. Let us know if you have other suggestions here.
>
> > [T]he flow of the paper is somewhat confusing as, e.g., the term "neuron" is core since page 2 but it is defined/explained in page 7.
>
> We appreciate this comment; it came up while we were writing the paper. For organizational reasons, we focus more on defining causal graph terminology early in the paper, and keep the framework intentionally abstract at first. We decided to define neural network component types such as mediators and submodules closer to Section 4, when we begin precisely describing and comparing different mediator types; we believe having these technical definitions closer to where they are more relevant will make referencing them easier. We have refactored the paper to make it clear when this information will become relevant, and to advise readers earlier in the paper where they can find this information.

---

### Review · Reviewer_EVLS · 2025-01-02

**Summary Of Contributions:**

The paper is concerned with causal interpretability where the goal is to extract a causal graph from trained neural nets.

The authors make a good case for the need for more foundational work in XAI. In the field of neurosymbolic AI, logical expressions are normally extracted from trained networks, sometimes in the form of decision trees or graphs or logical implications. Such implication rules can be chained when the network is decomposed as part of the rule extraction process. The fidelity of such rules can be measured against the behaviour of the network, offering a metric that goes beyond specific dataset evaluations. Such implication rules are probably closest to the causal graphs that are investigated here.

There's a gap however in the literature review which jumps from early work directly to LIME and SHAP and the latest LLM-based models without reference to relevant work in the area of neurosymbolic AI and knowledge extraction. For example, concept bottlenecks and layerwise knowledge extraction seem to be relevant to finding good candidate mediators, as well as the idea of network probing in order to obtain insight on possible mediators:

https://arxiv.org/pdf/2003.09000

https://arxiv.org/abs/1711.11279

http://www.interpretable-ml.org/nips2017workshop/papers/12.pdf

Of particular relevance in connection with the critique of LIME and SHAP is the measurable counterfactual approach below:

https://arxiv.org/abs/1908.03020

**Audience:**

Yes

**Claims And Evidence:**

Yes

**Requested Changes:**

The key question of course is the choice of mediators and the computational complexity of the proposed process.

In what concerns the survey, I'd have expected to see a reference made to the differences between layerwise, global and local XAI approaches.

Another fundamental question relates to whether an assumption can really be made that the relationships extracted from trained networks are truly causal other than in the much more limited sense that activating certain neurons "cause" activation of certain other neurons.

The paper states: "while mechanistic understanding is frequently discussed, the basic causal units underlying these mechanisms are often not explicitly defined". This discussion would benefit from a precise definition as to whether "causal" is intended in the broader or the above narrow sense that can be measured with a fidelity metric, as done in:

https://link.springer.com/article/10.1007/s10994-023-06333-w

Minor:

The figures are not very informative and could be improved to convey the specific message of the paper.

**Strengths And Weaknesses:**

The proposal to identify "particular kinds of mediators that are most appropriate depending on the goals of a given study" is a good idea and it should be pursued. The problem is that the paper lacks much indication or example of what this may look like in practice. Perhaps while seeking to survey the history, propose a taxonomy and investigate the value of mediators, the paper falls short in each of the areas. The authors could consider having a focus on either the survey and taxonomy or the specific mediator algorithms with examples and evaluation metrics. Arguing for a "more cohesive narrative of the field, as well as actionable insights" is one thing. Discovering "new mediators and sophisticated abstractions from neural networks than the primarily linear mediators employed in current work" is quite another - and it's an important part of the research in neurosymbolic AI.

---

> ### Author Response · Authors · 2025-01-10
> **Response to EVLS**
>
> Thank you for the thoughtful comments.
>
> > There's a gap [...] in the literature review[...].
>
> Thank you for bringing these papers to our attention. We have added most of the linked XAI papers to the History section. We have a full section on network probing (Section 5.2.1). We have added references to concept bottleneck models in the Discussion when referring to models trained to have inherently interpretable components.
>
> > [T]he paper lacks much indication or example of what this may look like in practice.
>
> Section 7.1 is dedicated to this topic. In this section, we ask: given a particular research goal, what kinds of mediators are best? To summarize, fine-grained mediators, such as autoencoder features or neurons, are best if one’s goal is to understand how a model accomplishes a task. If one’s goal is merely to figure out which parameters to prune or modify to improve performance, more coarse-grained mediators such as layers and submodules are acceptable. Time permitting, we would appreciate elaboration on this point!
>
> Thank you for the suggestion to add concrete examples. We have added example research scenarios to Section 7.1.
>
> > …consider having a focus on either the survey and taxonomy or the specific mediator algorithms with examples and evaluation metrics.
>
> We appreciate this suggestion, and see the motivation for this feedback. We have chosen to keep both: mediator search methods are inextricably linked to mediator types, and reflect fundamental strengths and weaknesses of these mediator types. Thus, we consider Section 5 highly relevant to Section 4, and not an orthogonal survey. For example, a discussion of the pros and cons of neurons versus non-basis-aligned directions would not be complete without mentioning that the latter cannot be as easily searched over without external trained components.
>
> We acknowledge that this link was not clear. We have added Table 1 and a more detailed Discussion to address this.
>
> >  I'd have expected to see a reference made to the differences between layerwise, global and local XAI approaches.
>
> Thank you for raising this point. Much of Sections 4 and 5 are concerned primarily, though not exclusively, with global XAI. We wish to understand why a model behaves in a particular way across examples, such that we can predict future successes and failures. (That said, we cite local XAI methods like LIME in Section 3.) Technically, these methods could also be applied to understand local predictions, but general understanding is more often the goal in contemporary work. We have added a note on this.
>
> > [Can] an assumption [] really be made that the relationships extracted from trained networks are truly causal other than in the much more limited sense that activating certain neurons "cause" activation of certain other neurons[?].
>
> Yes! The target metric is often defined such that we only locate mediators that have a measurable impact on the final model behavior. This includes edges: works like [1] and [2] define the importance of an edge *not* w.r.t. whether it causes a downstream neuron to activate, but instead w.r.t. where information pertinent to *the final model behavior* flows. For example, we might only include an edge between two neurons if removing this edge directly and measurably causes a change in the output behavior.
>
> > This discussion would benefit from a precise definition as to whether "causal" is intended in the broader or the above narrow sense that can be measured with a fidelity metric[...].
>
> By “cause”, we generally mean a global counterfactual influence. Thus, a cause should be included in our causal graph only if it is likely to have a measurable influence on what a model does across examples in a given task setting. This is quantified via the indirect effect (IE). This metric does resemble the cited fidelity metric, though IE is more often used to quantify counterfactual influence across examples, and is less concerned with explaining individual model predictions (though could be used for this purpose!). We have added a note about this in the History section.
>
> > The figures are not very informative and could be improved to convey the specific message of the paper.
>
> Time permitting, could you elaborate? Figure 2 illustrates the causal mediation analysis framework, Figure 3 visualizes the types of mediators in our survey, and Figure 4 illustrates the intuition underlying non-basis-aligned mediators. We believe these help demonstrate important concepts that are not as easily conveyed in text. If you have specific suggestions as to how these could be improved, we would love to improve our paper!
>
> References
> ===
> [1] Michael Hanna et al. (2024). “Have faith in faithfulness: Going beyond circuit overlap when finding model mechanisms.” COLM. https://openreview.net/forum?id=grXgesr5dT
>
> [2] Samuel Marks et al. (2024). “Sparse feature circuits: Discovering and editing interpretable causal graphs in language models.” https://arxiv.org/abs/2403.19647

---

### Author Response · Authors · 2025-01-10
**Global response**

We thank the reviewers for their constructive feedback. We have uploaded a revised version of the PDF with changes highlighted in purple. Changes include the following:

* We have expanded our discussion of when certain mediators are better than others (Section 7.1), and added concrete examples of what the process of picking a mediator looks like.
* A summary figure (Figure 1), survey summary table (Table 1), and more cross-references to aid navigability.
* A more detailed discussion of causal preliminaries in Section 2.
* More principled formal notation.
* Better transitions and more direct links to our main claims throughout the paper. Specifically, we have added more direct references throughout the paper to our claim that the perspective we present is novel and encourages focus on new and important lines of research.
* We have more explicitly contrasted this survey with existing work in the Related Work section, and expanded our discussion of interpretability tools.
* Small revisions to existing figures: Figure 3 (formerly Figure 2) now has a “submodule” label. Figure 4 (formerly Figure 3) uses updated notation that distinguishes between neurons and their activations.

We address specific points raised by each reviewer in individual responses.

---

### Decision · Action_Editor_KH9d · 2025-03-10

**Recommendation:** Reject

**Comment:**

While the authors have made progress in addressing the reviewers' concerns in their revision, the consensus among the reviewers is that the submission is not yet above the acceptance threshold. For instance, Reviewer J7Fd thought that their requests for clearer and more precise definitions of abstract concepts were not fully integrated into the paper. As an example, they pointed out that while the authors added a brief definition of a causal graph, they didn't consistently use the introduced notation throughout the paper.

I encourage the authors to perform a more thorough revision, fully incorporating the reviewers' suggestions, and then consider resubmitting.

**Audience:**

The reviewers agree that this is an important area of research that spans multiple fields, including causal inference, neurosymbolic AI, and interoperability. They agree that a well-structured survey would be valuable in organizing this literature and serving as an entry point for researchers.

**Claims And Evidence:**

This submission provides a survey on casual interpretability, focusing on extracting causal graphs from trained neural networks. The authors are not making claims of novelty.

**Resubmission Of Major Revision:**

The authors may consider submitting a major revision at a later time.